# Response to comment on 'Criterion placement threatens the construct validity of neural measures of consciousness'

**Johannes Jacobus Fahrenfort[1,2,3,4]\*, Philippa A Johnson[5], Niels Kloosterman[6,7], Timo Stein[3,4], Simon van Gaal[3,4]**

[1]Department of Applied and Experimental Psychology, Vrije Universiteit Amsterdam, Amsterdam, Netherlands; [2]Institute for Brain and Behavior Amsterdam (iBBA), Vrije Universiteit Amsterdam, Amsterdam, Netherlands; [3]Department of Psychology, University of Amsterdam, Amsterdam, Netherlands; [4]Amsterdam Brain and Cognition, University of Amsterdam, Amsterdam, Netherlands; [5]Cognitive Psychology Unit, Institute of Psychology and Leiden Institute for Brain and Cognition, Leiden University, Leiden, Netherlands; [6]Department of Psychology, University of Lübeck, Lübeck, Germany; [7]Center of Brain, Behavior and Metabolism, University of Lübeck, Lübeck, Germany

**\*For correspondence:**
j.j.fahrenfort@vu.nl

**Competing interest:** The authors declare that no competing interests exist.

**Abstract** In Fahrenfort et al., 2025 we show the influence of non-perceptual criterion shifts on neural measures of consciousness. We fully agree (and point out in our article) that it was already known that subjective measures are sensitive to criterion confounds, and we are happy to read that this is acknowledged by Sandberg and Overgaard in their comment (Sandberg and Overgaard, 2025). However, we contest that the main findings of our simulations and empirical studies had already been demonstrated. Several findings from our studies are novel, such as the fact that criterion effects reveal themselves as over- (or under-) estimations of both conscious and unconscious processing in tandem, and that this has tangible implications when analyzing real neural data. We also challenge the suggestion that our experimental manipulations are (too) radical compared to signal-to-noise variations that occur naturally between experiments.

## Assumptions of subjective measures

In *Fahrenfort et al., 2025* we state that the underlying assumption is that selecting [0] will only occur if trials are 'truly' unseen. This statement is derived from the description of the PAS itself, stating that [0] refers to either: "No experience. No impression of the stimulus is experienced. All answers are experienced as mere guessing." (*Overgaard et al., 2006*), "No experience" (*Sandberg et al., 2010*), "No experience: No subjective experience of the stimulus, not even the 'faintest sensation' that anything was presented at all. Not even a feeling that something might have been presented" (*Overgaard and Sandberg, 2021*). The implication of accepting the validity of this category is clearly that it measures what the label refers to, that the stimulus is 'truly' not experienced (although the exact phrasing of this category may vary slightly between studies). This indeed is how it is used in the literature, for example when making claims about the existence of unconscious working memory, and many other claims regarding unconscious processing (see references in our manuscript).

Sandberg and Overgaard now claim that it is a niche position to suggest that "awareness ratings should be treated as flawless insights into participants' experience", referring to their own care in

highlighting that this implication is unwarranted (*Sandberg and Overgaard, 2025*). They further underpin this by reference to the fact that 32 researchers advise against relying on subjective measures alone to establish evidence of unconscious processing (*Stockart et al., 2024*, p. 14). Indeed, wise advice that is supported by our manuscript, although we would like to note that a relatively recent survey has shown that most consciousness researchers believe that subjective measures - and the PAS specifically - are the best measures to check whether a stimulus is consciously perceived (*Francken et al., 2022*, Figure 4). We fail to see how highlighting imperfections of subjective measures can serve as an argument in their defense, or how these concerns disappear when acknowledging them. Similarly, we fail to see how the continued use of subjective measures (with or without the acknowledgement of their imperfection) builds support for their continued use.

## Similarity of findings – different conclusions

A second criticism of Sandberg and Overgaard is that 'the main findings <of our manuscript> have already been demonstrated in other experiments'. To support this idea, they reference a publication (*Sandberg et al., 2022*) in which they model behavioral data from subjective reports. We have trouble matching the results presented in that article to the data, simulations and analyses from our own publication. Their article is not about neural measures of consciousness, nor does it contain brain imaging data. As such it bears little resemblance to our studies. Our article is specifically about the effect criterion shifts have on post-hoc sorted data to create neural conditions that are claimed to correspond to 'real' subjective states (such as expressed in the labels of the PAS, or in a 'yes'/'no' response). Further, we use empirical data from two EEG experiments to support the claim that the effects that we model in a signal detection framework are not merely theoretical artifacts but can have real implications for claims regarding the neural correlates of consciousness (NCC). The only correspondence seems to be that we agree on the somewhat unreliable nature of the PAS response, when discussing the fact that the PAS is not exhaustive (i.e. it may or may not capture weak conscious experiences depending on the experimental context).

Further, Sandberg and Overgaard claim that they had already established that report criteria depend on experimental context (*Skewes et al., 2021*). However, that article is intrinsically very different from ours, because it provided false performance feedback, which also affected participants' accuracy. In our experiments we only provided veridical feedback and specifically kept accuracy the same between conditions so that the effects are criterion specific and not related to general effects of sensitivity or other cognitive effects. More importantly, the aim of our paper was not to show that report criteria depend on experimental context. Rather, we took this as a starting point to show the effect of this contextual change on neural correlates based on subjective measures and post-hoc sorting, including the PAS. Furthermore, we not only show that the experimental context influences report criteria, which should be well-known (although we contest that this is widely accepted given the lenient use of subjective measures in the literature), but we also show through simulation that the experimental context determines whether the relative confounding effect of criterion placement is larger in neural measures of either conscious or unconscious processing.

## Assumptions of instructions

Next, Sandberg and Overgaard contend that we made changes to the PAS that we ourselves consider so substantial that it may be argued that we did not use the PAS at all (referencing our Discussion). However, this is not what we argued. Rather, we merely acknowledged Sandberg and Overgaard's potential concern on our usage of the PAS, already in the first version of the manuscript prior to three peer reviews. Subsequently, after personal communication with them, we conceded that a particular sentence that Sandberg and Overgaard highlight and that occurred somewhere in the instructions ("Only press 0 if you are 100% convinced that no square appeared and only press 3 if you are 100% convinced that a square appeared.") might be misconstrued by participants as a general confidence rating. To give them a fair hearing in our article, we expounded on this issue in an updated Discussion, and we asked them beforehand whether they agreed with the way we explicitly highlight this concern in the discussion of our Version of Record (to which they agreed).

However, this does not mean that we believe that our usage of the scale should not be characterized as the PAS. The PAS refers to verbal labels of a scale, i.e. it is a measurement instrument. In our

experiment, the proper PAS labels were used throughout the experiment and repeatedly shown to remind participants of what we specifically asked of them. Thus, we disagree with the statement that we "did not use the PAS at all" and we do not believe that the single sentence that Sandberg and Overgaard highlighted changes this fact or would have meaningfully changed the outcome of the study.

The other criticism is that we used punishments in our experiment, and that "punishment can be used to disrupt the result of essentially any psychological test". This might be true if we had provided participants with false feedback, which we did not. We only gave veridical feedback to experimentally create criterion shifts, the same shifts that Sandberg and Overgaard acknowledge also occur naturally in other contexts given their admission that the PAS is not an exhaustive measure. We made explicit that our criterion manipulation was strong ("As such, the current experiment can be viewed as a caricature of actual experimental practice", page 13 top), but we disagree that such changes do not occur across experimental contexts, or even that our experimental context is highly unusual (see references in our manuscript). Indeed, one can observe similar criterion shifts in experiments comparing experimental blocks with different base-rates of stimulus occurrence (e.g., more vs less targets) without punishment manipulations (*Sánchez-Fuenzalida et al., 2025*; *Sánchez-Fuenzalida et al., 2023*). These studies also reveal that such criterion shifts do not affect conscious experience. If a measurement instrument is strongly affected by naturally occurring contextual variations, we believe it is reasonable to use experimental manipulation to show how these confounds can materialize in real data.

## Conclusions

Concluding, we do not agree that our study primarily adds detail to what was already known about the limitations of subjective measures, as we explain here and in the manuscript. We also contest that our criticisms can be mitigated simply by acknowledging limitations of subjective measures. Claims based on post-hoc sorting of subjective reports are not likely to agree across experimental contexts, so that conclusions regarding the depth or extent of unconscious processing, or regarding the temporal or spatial profile of the NCC, vary considerably across the literature (*Yaron et al., 2022*). With respect to subjective measures, this is not due to mere 'limitations', but due to a serious confound in neural activation patterns based on subjective measures, one that deserves considerable attention. We hope our manuscript contributes to raising awareness about this confound.

## Additional information

### Funding

| Funder | Grant reference number | Author |
|---|---|---|
| HORIZON EUROPE European Research Council | 10.3030/715605 | Simon van Gaal |

The funders had no role in study design, data collection and interpretation, or the decision to submit the work for publication.

### Author contributions

Johannes Jacobus Fahrenfort, Writing – original draft, Writing – review and editing; Philippa A Johnson, Niels Kloosterman, Timo Stein, Writing – review and editing; Simon van Gaal, Funding acquisition, Writing – review and editing

### Author ORCIDs

Johannes Jacobus Fahrenfort https://orcid.org/0000-0002-9025-3436
Philippa A Johnson https://orcid.org/0000-0002-6125-3138
Niels Kloosterman https://orcid.org/0000-0002-1134-7996
Timo Stein https://orcid.org/0000-0002-8484-0933
Simon van Gaal https://orcid.org/0000-0001-6628-4534

**Decision letter and Author response**

Decision letter https://doi.org/10.7554/eLife.107622.sa1

**Data availability**

This manuscript does not contain data.

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
