## [Decision Letter]

*In the interests of transparency, eLife includes the editorial decision letter. A lightly edited version of the letter sent to the authors after peer review is shown, indicating the most substantive concerns; minor comments are not usually included. Since no essential revisions were requested there is no accompanying author response.*

Thank you for submitting your article 'Response to comment on 'Criterion placement threatens the construct validity of neural measures of consciousness' to *eLife*. Your article has been reviewed by two peer reviewers, and the evaluation has been overseen by a Reviewing Editor (Ming Meng) and a Senior Editor (Joshua Gold).

Reviewer Comments

*Reviewer #1:*

This response effectively defends the original paper's novelty and methodological rigor against critiques by Sandberg and Overgaard. It robustly clarifies the core findings. They found that criterion shifts confounding neural measures of consciousness using simulation and EEG measurements. The authors insist PAS [0] should reliably isolate unconscious processing if instructions are followed. Critics counter that no subjective measure can objectively validate the absence of consciousness, making neural sorting inherently confounded. The debate remains unresolved because it hinges on irreconcilable assumptions about the reliability of introspection and the ethics of experimental design. The authors' empirical demonstrations (using EEG) strengthen their case. Overall, the reply strengthens the original argument that neural data sorted by subjective reports are vulnerable to contextual confounds, urging methodological caution in consciousness research.

Substantive concerns

1. The debate exposes a field-wide impasse: If all measures of consciousness rely on subjective reporting (e.g., PAS), and subjective reports are inherently criterion-dependent, how can neural correlates of consciousness (NCC) ever be empirically validated? The authors treat PAS [0] as a "ground truth" for unconsciousness, while critics argue this category is irreducibly contaminated by metacognitive biases.

2. The debate raises the concern that whether we should choose between naturalistic observation (preserving ecological validity) and strong manipulations (exposing measurement flaws).

3. The disagreement of PAS [0] stems from deeper divides about whether consciousness can exist independent of metacognitive access-a problem neuroscience cannot resolve without philosophical engagement.

4. The authors demonstrate that criterion shifts asymmetrically bias neural "conscious" vs. "unconscious" conditions. This challenges the field's pursuit of universal NCC markers, suggesting consciousness signatures may be methodological artifacts rather than stable biological phenomena.

*Reviewer #2:*

The response effectively addresses the queries raised in the comment article by clarifying the core assumptions of the original study, highlighting the main findings, and refuting the criticisms with evidence. It provides detailed explanations for the valid concerns raised in the comment and justifies why the study's main hypotheses and conclusions remain unaffected by the use of PAS and punishment for criterial modulation.

The authors are encouraged to address the section on Assumptions of subjective measures and acknowledge the critique that PAS may not elicit purely unconscious processing (0.9 rather than 0.8, but not 1). Despite this acknowledgment, the key point remains that the original study's findings suggest that due to criterion shifts, neural decoding analyses relying on PAS behavioral reports during sorting trials would be biased. This bias is driven by criterion shifts and SNR, rather than the reporting method (PAS or non-PAS).